# BARE: Leveraging Base Language Models for Few-Shot Synthetic Data Generation

## Abstract

As the demand for high-quality data in model training grows, researchers and developers are increasingly generating synthetic data to tune and train LLMs. However, current data generation methods rely on seed sets containing tens of thousands of examples to prompt instruction-tuned models. This reliance can be especially problematic when the large-scale collection of high-quality seed examples is expensive or difficult. In this paper we explore the novel few-shot synthetic data generation setting – generating a high-quality dataset from only a few seed examples. We show that in this low-seed setting, instruction-tuned models used in current synthetic data methods produce insufficient diversity for downstream tasks. In contrast, we show that base models without post-training, largely untapped for synthetic data generation, offer substantially greater output diversity, albeit with lower instruction following abilities. Leveraging this insight, we propose **Base-Refine** (BARE), a novel two-stage method that combines the diversity of base models with the quality assurance of instruction-tuned models. BARE excels in few-shot synthetic data generation: using **only 3 seed examples** it generates diverse, high-quality datasets that significantly improve downstream task performance. We show that fine-tuning Llama 3.1 8B with 1,000 BARE-generated samples achieves performance comparable to state-of-the-art similarly sized models on LiveCodeBench tasks. Furthermore, data generated with BARE enables a 101% improvement for a fine-tuned Llama 3.2 1B on GSM8K over data generated by only instruction-models, and an 18.4% improvement for a fine-tuned Llama 3.1 8B over the state-of-the-art RAFT method for RAG data generation.

## 1 Introduction

As Large Language Models (LLMs) grow in size and capability, the demand for high-quality, diverse data in model training is outpacing human-generated data, necessitating the use of synthetically generated data (Villalobos et al., 2024; Dubey et al., 2024; Qwen, 2025; Nvidia, 2024; Guan et al., 2025; NovaSky Team, 2025).

However, prevailing methods for synthetic data generation require developers to provide large and diverse seed sets as the first step in their generation pipeline. For instance, OSS-Instruct (Wei et al., 2024) uses 80,000 code snippets and Humpback (Li et al., 2024c) over 500,000 text segments. This reliance on large-scale manually curated seed data imposes significant burdens on developers. For example, consider a specialized setting like course-specific grading (Latif & Zhai, 2024), where an instructor would like to have models automatically grade essays. Here, collection of seed data would be difficult and costly, requiring thousands of manually graded essays.

Large seed sets are necessary in current pipelines to ensure **data diversity**, a key property of effective synthetic data (Chen et al., 2024; Raventós et al., 2023). Recent work suggests that instruction-tuned LLMs suffer from a lack of diversity due to the post-training process, where standard techniques lead to mode collapse (Shumailov et al., 2024; Wong et al., 2024; Lambert et al., 2024), limiting a model's ability to generate varied responses to open-ended queries. Additionally, we show that methods to induce diversity via prompting (e.g. including past examples in context) (Zhang et al., 2024a; Naik et al., 2023; Fröhling et al., 2024) are insufficient for few-shot generation. *Thus, a critical need exists for methods that can generate diverse and high-quality datasets from minimal examples.*

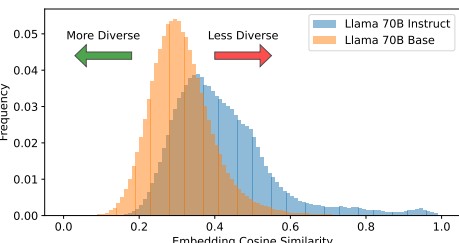 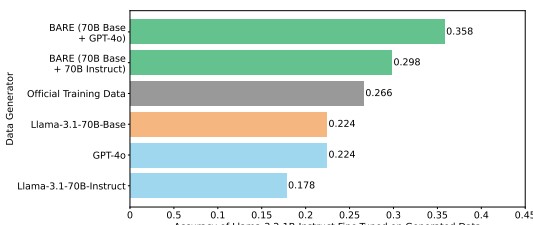

Figure 1: **Histogram of pairwise embedding cosine similarity scores** for 1000 Llama-3.1-70B-Base vs Instruct generations of grade school math problems with 3 seed examples. The base distribution is further left, indicating lower similarity and hence higher diversity.

Figure 2: **Accuracy of a Llama-3.2-1B-Instruct model** fine-tuned on real-world data and 5 different sets of math problems synthetically generated using 3 seed examples, evaluated on GSM8K. Training with BARE-generated data outperforms all other data sources.

**Base Models for Diversity.** Instead of large seed sets, an underexplored alternative source of diversity is the use of base models. Not being subject to the same post-training procedures, base models better reflect the diversity found in real-world data (OpenAI, 2024b). Intuitively, this allows practitioners to achieve diversity by leveraging the underlying model of language in base models rather than spending the effort to collect large-scale seed sets. Quantitatively, base models do demonstrate greater diversity. As shown in Figure 1, they generate outputs with noticeably lower pairwise embedding cosine similarity (mean 0.313) compared to instruction-tuned models (mean 0.421); lower cosine similarity indicates greater diversity. This increased diversity can benefit downstream tasks: Figure 2 shows that fine-tuning Llama-3.2-1B-Instruct on GSM8K using data from Llama-3.1-70B-Base yields a 22.5% accuracy, compared to 17.8% using Llama-3.1-70B-Instruct.

However, due to their weaker instruction-following capabilities, base model generations often suffer from lower quality (Figure 4), which can negate the benefits of increased diversity and hinder downstream training performance. Our key insight is that by managing the quality of individual data points using instruction-tuned models, we can harness the diversity advantage of base models.

**BARE.** Leveraging the distinct strengths of base and instruction-tuned models, we introduce **Base-Refine** (BARE): a simple yet novel approach for few-shot synthetic data generation. BARE leverages a base model to generate diverse initial generations with minimal seed data and refines each initial generation according to desiderata with instruction-tuned models. In a variety of few-shot settings, we show our two-stage process enhances diversity without compromising quality, enabling the generation of datasets that improve downstream performance when **only 3 seed examples** are available. *This is the first work to show the value of base models for data generation.*

On the LiveCodeBench Test Output Prediction task, fine-tuning with 1,000 BARE-generated examples achieves performance comparable to state-of-the-art models of similar size. Further, when applied to the Retrieval-Augmented Fine-Tuning (RAFT) method (Zhang et al., 2024b)—a state-of-the-art technique for generating synthetic Q&A pairs over retrieved documents for RAG—replacing RAFT's instruction-tuned only generator with BARE improves the fine-tuned model's accuracy by up to **18.4%** over the original RAFT approach (Figure 8). In mathematical reasoning, as exemplified by synthetic GSM8K-style problems, BARE using Llama-3.1-70B-Base for initial generation and GPT-4o for refinement with BARE increases a fine-tuned Llama-3.2-1B-Instruct model's accuracy on GSM8K to 35.8%, exceeding the 22.4% using instruct-only generation with GPT-4o and the 26.6% training on examples from the official human-generated GSM8K training set (Figure 2).

To summarize, our contributions are:

1. We quantitatively investigate the quality and diversity of base and instruction-tuned models across various sampling methods to motivate better system design. We show that base models tend to produce far more diverse responses whereas instruction-tuned models offer higher quality.

2. Building on these insights, we propose **Base-Refine** (BARE), a practical method novelly leveraging base models for the under-explored task of few-shot synthetic data generation. We demonstrate that BARE consistently outperforms past methods on small seed sets, including SOTA genera-

tion methods. Generating from as few as 3 seed examples, `BARE` can yield fine-tuned models comparable to SOTA models of similar sizes trained with much larger seed sets.

## 2 RELATED WORK

**Synthetic Data.** Synthetic data is commonly used to train state of the art models like Llama 3.3 (Dubey et al., 2024), Qwen 2.5 (Qwen, 2025), Nemotron 4 (Nvidia, 2024), and o3-mini (Guan et al., 2025). However, prior work has shown its usage poses risks such as model collapse, where iterative training on low-diversity data shifts the generation distribution toward a high-probability mean, degrading both performance and diversity (Shumailov et al., 2024; Shimabucoro et al., 2024; Guo et al., 2023). Recent work indicates that diversity in training data can improve downstream performance (Chen et al., 2024). However, this research often does not simultaneously consider the quality of the data in tandem. In contrast, the `BARE` pipeline is designed with the twin objectives of diversity and quality in mind, producing diverse, high-quality data to support model training.

**Generation Methods.** Sampling methods like temperature scaling and nucleus sampling (Holtzman et al., 2020) are widely used to improve diversity, but often prioritize token-level randomness over semantic diversity. Indeed, methods such as logit suppression (Chung et al., 2023) can enhance diversity but may require significant manual refinement to maintain quality unlike `BARE` which maintains quality as a first order requirement.

Many current synthetic data generation processes instead rely on varying the prompt to elicit a diverse dataset. One common approach is to rely on heavily curated prompts limiting scalability (Zhang et al., 2024a; Naik et al., 2023; Fröhling et al., 2024; Chen et al., 2024; Li et al., 2022). Another effective approach is to utilize a large diverse seed set, up to 100,000s of examples in size (Lambert et al., 2024; Li et al., 2023; Wei et al., 2024; Li et al., 2024c), requiring significant curation effort. `BARE` obviates both of these concerns by focusing on the few-shot setting (we use only 3 seed examples) and showing significant performance gains without prompt engineering.

Others have explored using an LLM to generate the seed set (Wong et al., 2024; Li et al., 2024a), but still use instruction-tuned models. These methods thus still suffer from the limited diversity of instruction-tuned models. In contrast, `BARE` utilizes base models to generate a large and diverse seed set. To our knowledge, no prior work has focused on this approach. Though prior work has studied differences in base and instruct models in calibration (OpenAI, 2024b) and agentic environments (Li et al., 2024b), no work has leveraged base models for synthetic data.

**Evaluating Synthetic Data.** The utility of synthetic data is typically assessed by downstream performance. However, to motivate the development of better systems, we also study diversity and the quality of individual entries (entry-wise quality). Token-level metrics like self-BLEU (Zhu et al., 2018) are common, though embedding-based approaches that are commonly used (e.g., BERTScore (Zhang et al., 2019), Sentence-BERT (Reimers & Gurevych, 2019)) better capture semantic diversity rather than token diversity, which is our primary focus. Lastly, while dataset-wide quality is often measured via downstream performance, assessing individual synthetic samples remains underexplored. In Section 3, we introduce an entry-wise quality measure to evaluate sample realism, ensuring robust synthetic data generation.

## 3 MOTIVATION

Synthetic data generation should result in a diverse and high-quality dataset. In this section, we run experiments to demonstrate the challenges of few-shot synthetic data generation with current methods and the differences between base and instruction tuned models. We demonstrate the potential of base models as a source of diversity, motivating the design of `BARE`.

### 3.1 DIVERSITY & QUALITY METRICS

**Diversity.** Following Tevet & Berant (2021); Cann et al. (2023); Cox et al. (2021), we use the average neural similarity score to measure the diversity of a generated dataset. Specifically, we use OpenAI's `text-embedding-3-small` (OpenAI, 2024a) to generate embeddings and use cosine

similarity to calculate similarity scores, as recommended by OpenAI. We calculate pairwise cosine similarity scores for items in a generated dataset and analyze the resulting distribution of similarities. A lower average similarity indicates a more diverse dataset.

**Entry-wise quality.** To measure entry-wise quality, we propose the *indistinguishability rate* (IR), a novel metric inspired by the adversarial framework of Generative Adversarial Networks (GANs) (Goodfellow et al., 2014). We task a strong LLM (e.g., GPT-4o) as a "discriminator" to distinguish a synthetic entry from $n = 3$ real entries. The IR is the rate at which it fails to distinguish the synthetic entry. A high IR suggests the synthetic data closely mimics real data, while a low IR indicates it is easily identifiable as out-of-distribution. An example IR prompt is in Appendix D.

## 3.2 EXPERIMENTAL SETUP

**Models.** To investigate diversity and quality differences between instruction-tuned and base models, we evaluate synthetic datasets generated using Llama-3.1-70B-Instruct and Llama-3.1-70B-Base (Dubey et al., 2024). We also use GPT-4o (OpenAI, 2024c) as a stronger instruction-tuned model.

**Domains.** We enforce the few-shot setting by exposing each generation method to only three real-world examples, simulating low seed data collection effort. We evaluate in the following established benchmarks:

- **Enron Emails** (Klimt & Yang, 2004) generating training data for classifying emails as spam or legitimate. We ensure class-balanced synthetic data by explicitly conditioning each generation on a uniform class distribution.
- **20 Newsgroups** (Pedregosa et al., 2011) generating training data for classifying Usenet messages into one of 20 newsgroup sources. We generate classes along with content, allowing the generation method to determine the class distribution of the synthetic dataset.
- **Retrieval-Augmented Fine-Tuning** (RAFT) (Zhang et al., 2024b), a domain and generator-agnostic synthetic data generation framework for fine-tuning data in RAG tasks. Q/A pair generation is conditioned on contexts mimicking retrieval results from a corpus. We use:
  - **HotpotQA** (Yang et al., 2018), a general Wikipedia-based short-answer task.
  - **PubMedQA** (Jin et al., 2019), a medical abstract-based yes/no/maybe question-answering task.
- **GSM8K** (Cobbe et al., 2021), generating grade-school math problems and solutions for fine-tuning.
- **LiveCodeBench's Test Output Prediction** (LCB TOP) (Jain et al., 2024), generating coding questions and answers on predicting test case outputs given a natural language description of an algorithm's expected behavior and test input for fine-tuning.

For the classification tasks of Enron and Newsgroups, we generate a dataset of size $n = 500$. For the generative model fine-tuning tasks of HotpotQA, PubMedQA, GSM8K, and LCB TOP, we generate a larger dataset of size $n = 1000$. For the RAFT domains, this would mean 10 questions for each of 100 simulated retrievals; we do not compute diversity for RAFT due to differences in retrievals.

We use straight-forward prompts with limited domain-tailoring (examples in Section D). Further sampling details, such as temperature, are in Section A.

## 3.3 INVESTIGATION RESULTS

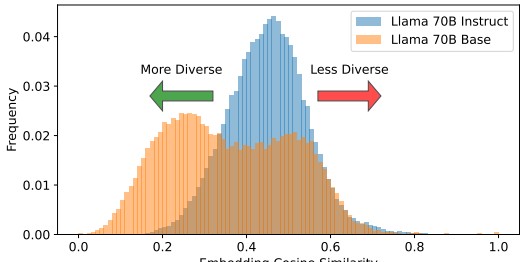

Figure 3: **Distribution of pairwise embedding cosine similarity scores** for Llama-3.1-70B-Instruct and Llama-3.1-70B-Base generations for Enron spam. Base model distribution have more density in the low-similarity region and less density in the high-similarity region, indicating greater diversity.

**Diversity.** From the pairwise cosine similarity distributions of the embeddings in Figure 3 (and recalling Figure 1), we see the base distribution is consistently further to the left, indicating that the base model generations are more diverse. The better diversity of base models is further reflected

Table 1: **Average pairwise embedding cosine similarity of Llama-3.1-70B-Instruct vs.** Llama-3.1-70B-Base generated data. Generations from Base are almost always more diverse than Instruct, despite always sampling at a lower temperature (0.7 vs 1.0).

|  | ENRON | NEWSGROUPS | GSM8K | LCB TOP |
|---|---|---|---|---|
| INSTRUCT | 0.450 | 0.256 | 0.421 | **0.389** |
| BASE | **0.350** | **0.162** | **0.313** | 0.468 |

Table 2: **Average pairwise embedding cosine similarity of various prompting techniques.** GPT-4o is used as the prompting generator due to the need for strong instruction following capabilities; Llama models frequently derailed. Llama-3.1-70B-Base generations are almost uniformly more diverse than any prompting method, except for persona prompting on GSM8K, where it is comparable.

|  | PROMPTING GPT-4O | | | | | LLAMA-3.1-70B-BASE |
|---|---|---|---|---|---|---|
|  | IND. | PERSONA | SEQ. | IN-ONE | DYN. FEWSHOT | INDEPENDENT |
| ENRON | 0.574 | 0.580 | 0.511 | 0.363 | 0.511 | **0.350** |
| GSM8K | 0.427 | **0.308** | 0.398 | 0.347 | 0.463 | *0.313* |

in Table 1, with all domains except one showing lower mean similarity for base models, indicating higher diversity. Upon inspection, we attribute the reversal in trend in LCB TOP to the base model repeating phrases from the examples. This is related to potential issues with the quality of base model generations, which we discuss below.

In addition to repeated independent sampling from the instruction-tuned model, we further explore diversity-eliciting prompting for data generation. We do not explore such methods on base models due to their poor instruction-following capabilities. The methods we examine are:

- **Persona Prompting**: Model responds as a predefined persona (Fröhling et al., 2024).
- **Sequential Prompting**: Model iteratively generates outputs distinct from previous ones.
- **In-One Prompting**: Model generates $k$ different entries in a single response (Zhang et al., 2024a).
- **Dynamic Few-Shot Examples**: Different few-shot examples are randomly selected for each call (using a larger seed set) (Li et al., 2022).

Table 2 shows that base models generally yield higher diversity than almost all prompting methods in the two domains, as measured by the average embedding distance. The exception is persona prompting on GSM8K — though this diversity arises more from flavor text differences due to personas rather than actual content.

We thus find that base models are generally more diverse than instruction-tuned models and that temperature increases and prompting methods are insufficient to bridge the gap, motivating our usage of base models in the first stage of BARE.

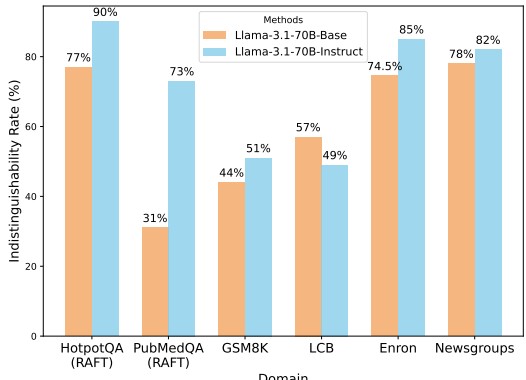

Figure 4: **Indistinguishability Rate for Llama-3.1-70B Base and Instruct data generation methods** across various datasets. Llama-3.1-70B-Instruct is almost uniformly better at generating examples that appear in-domain when compared alongside real-world data.

**Entry-wise quality.** Figure 4 presents our results. In general, the instruction-tuned model has a higher indistinguishability rate (IR) (see Section 3.1), indicating that it is better at producing generations that resemble high-quality data. Some IRs are well above 75%, which is not unexpected: if a model consistently generates data that aligns with the most common patterns in the real-world distribution, it becomes difficult

to distinguish from actual data. Since the real-world entries often also include non-modal (less frequent) samples, a discriminator tasked with identifying lower quality data may instead misclassify these less common real-world samples as synthetic.

As mentioned above, on LCB TOP the base model repeats phrases from examples in the prompt, leading to a higher IR as the repeated examples *are* real. Consequently, while individual base generations are technically more realistic, their shortcomings are captured in the diversity metric.

We thus find that the superior instruction following capabilities of instruction-tuned models generates more realistic data, motivating our usage of instruction-tuned models in the second stage of BARE.

## 4 BASE-REFINE (BARE)

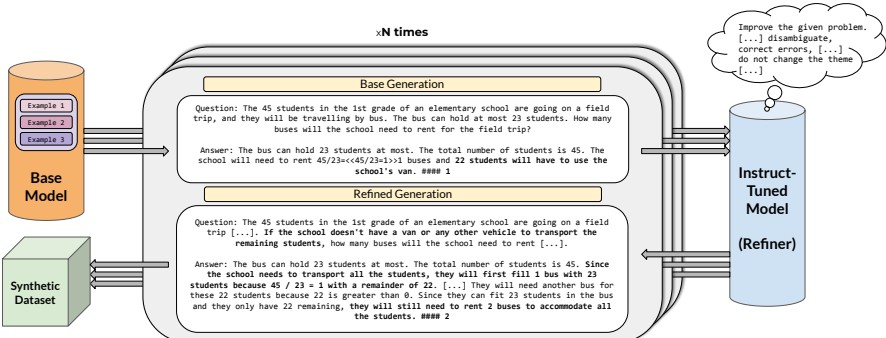

Figure 5: **BARE combines base models with instruction tuned models.** Instruction-tuned models provide high-quality but low-diversity data, while base models provide low-quality but high-diversity data. With minimal seed examples, BARE independently generates a diverse initial set of data points with a base model and refines each entry individually with an instruction-tuned model to create a high-quality, high-diversity dataset. In this example of a real grade school math problem generation, the Llama-3.1-70B-Base model hallucinates in its answer to its own question. The refiner (Llama-3.1-70B-Instruct) recognizes this and disambiguates the question and corrects the reasoning.

Building on the relative advantages of base and instruction tuned models we introduce BARE. BARE is a practical few-shot synthetic data generation method that combines the diversity of base models with the fidelity of instruction tuned models. BARE uses a base model to generate an initial set of diverse but potentially lower quality data from very little seed data. An instruction-tuned model then individually refines each example from the initial set, improving it according to specific criteria (e.g., realism, correctness) while keeping the original concept. The refinement retains the overall diversity of the initial set while exerting greater control over the quality of the final generations.

Importantly, diversity is elicited from the inherent properties of the base model rather than a large initial seed set, allowing for greatly improved data efficiency. Seed examples are only necessary to ensure the base model follows formatting (though they can also be included in the refine step). In our experiments, we use just three few-shot examples. In addition, we intentionally use very general prompts for BARE to demonstrate its flexibility, underscoring the potential for even greater improvement with tailored prompts. Representative prompts are included in Section D.

## 5 EVALUATION

We evaluate BARE on diversity, data quality, and downstream utility across the same domains and baselines presented in Section 3. BARE uses Llama-3.1-70B-Base for generation and Llama-3.1-70B-Instruct for refinement. We also experiment with the Llama 3.1 8B family and GPT-4o as the refiner (Dubey et al., 2024; OpenAI, 2024c). Sampling details are in Section A.

To evaluate the downstream utility of BARE, we LoRA (Hu et al., 2021) fine-tune Llama-3.1-8B-Instruct for 4 epochs using the generated data, except for GSM8K where Llama-3.2-1B-Instruct (Dubey et al., 2024) is fine-tuned instead due to high baseline performance of the 8B model. Other fine-tuning hyperparameters are in Section B.1. The fine-tuned models are evaluated on a static test

set for HotpotQA, PubMedQA, GSM8K, and LCB TOP. In this section, we will focus only on the generative tasks; the downstream evaluation process for classification tasks is presented in Section B.1 and detailed results for all domains can be found in Section B.2.

**Baselines.** To compare against existing synthetic data generation methods, our prompt for instruction-tuned models is comparable to those used in current synthetic data generation techniques such as OSS-Instruct (Wei et al., 2024). Indeed, validation training runs adapting OSS-Instruct's prompts to our setting yielded comparable results to our prompts. We thus use our instruction-tuned-only methods as a baseline representation for existing works that use instruction-models in their synthetic data generation from seed sets.

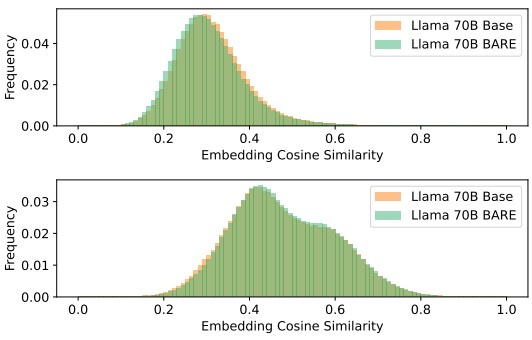

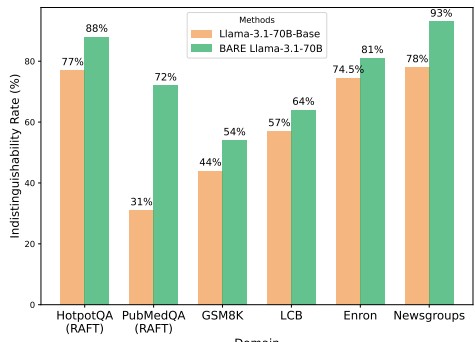

Figure 6: **Distribution of pairwise embedding cosine similarity scores** for Llama-3.1-70B-Base and Llama 3.1 70B `BARE` generations for GSM8K (top) and LCB TOP (bottom). The distributions are extremely similar for both tasks, indicating that refinement retains the diversity of base generations.

Figure 7: **Indistinguishability Rate** for data generated by Llama-3.1-70B-Base and `BARE` across various datasets. `BARE` consistently improves the quality of data generated from base models.

**`BARE` Quality & Diversity.** We begin by comparing the quality/diversity trends of `BARE` with other methods. From the histograms in Figure 6, we can see that `BARE` effectively does not change the similarity distribution of generated data when compared to the base model at all. Detailed results with average embedding similarity scores can be found in Section B.2.

At the same time, we see from Figure 7 that `BARE` leads to a monotonic increase in the IR for every domain - suggesting that it is able to lift the quality of the generations to be on par or in some cases even surpass directly sampling from an instruct model. Combined, the IR and diversity measures indicate that `BARE` is capable of leveraging the diversity of base models and quality of instruction-tuned models in its end generations.

**Fine-tuned Model Accuracy.** We now show the utility of `BARE`-generated datasets as a whole. In Figure 8, we demonstrate the accuracy of a model fine-tuned on datasets generated using different methods. `BARE`-generated data leads to better downstream models than data from either Base or Instruct models almost uniformly, and across all domains training with `BARE`-generated data leads to the highest model accuracy.

Surprisingly, in Figure 9, we find that `BARE` using the small Llama-3.1-8B base model to generate data and GPT-4o as a refiner, **outperforms** all advanced prompting methods discussed in Section 3.3 applied to GPT-4o directly. These clear results showcase `BARE`'s ability to out-perform existing methods for high-quality, diverse data generation in few-shot settings.

On RAFT domains, `BARE` improves upon the standard state-of-the-art pipeline for fine-tuning LLMs for RAG (RAFT) by up to 18.4%, as seen with the 8B family on HotpotQA (standard RAFT is represented in our Instruct generation results). `BARE` with both model families also outperforms existing RAFT pipelines on PubMedQA.

On GSM8K, `BARE` is the only method that provides useful training data. The un-trained model performance was 21.8%, and fine-tuning on `BARE`-generated data achieves accuracies of 24.9% and 32.8% with the Llama 8B and 70B families, respectively. Accuracy when training with data generated by single model methods either decreased accuracy or had little difference. In fact, training

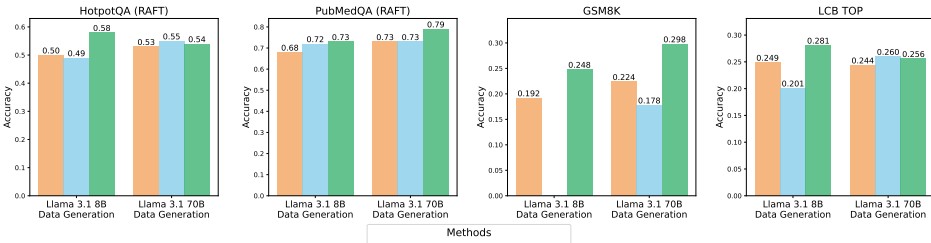

Figure 8: **Downstream accuracy of BARE compared to Llama-3.1 Base and Instruct models.** Accuracy is measured on Llama-3.1-8B-Instruct Model (HotpotQA, PubMedQA, LCB TOP) and Llama-3.2-1B-Instruct Model (GSM8K) finetuned on synthetic data generated using Base, Instruct, and BARE methods. Note that GSM8K 8B Instruct results are not shown as generations derailed. In general, we find that BARE out-performs directly using the base and instruction fine-tuned models.

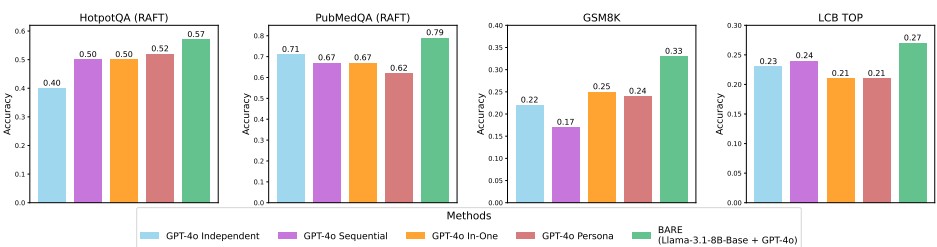

Figure 9: **Downstream accuracy for BARE versus other diversification methods.** Fine-tuned model accuracy on HotpotQA, PubMedQA, GSM8K, and LCB TOP for data generated from prompting methods using GPT-4o. Prompting methods have mixed effects on downstream performance when compared against standard sampling. Alternatively, BARE using just with the Llama-3.1-8B Base as the generator and GPT-4o as the refiner significantly outperforms all over methods using GPT-4o.

on data generated by Llama-3.1-70B-Instruct led to an accuracy of just 17.8%, which BARE with Llama-3.1-70B-Base refined by GPT-4o outperforms by over $2\times$ (35.8% accuracy).

On LCB TOP, fine-tuning a Llama-3.1-8B-Instruct model on 1,000 examples generated by BARE using the Llama 3.1 8B model family for just 4 epochs resulted in performance of 28.1% accuracy, comparable to the current top models of similar size on the LCB leaderboard: DeepSeekCoder 6.7B Instruct (Guo et al., 2024) at 32.7% and Magicoder$\mathcal{S}$ DS 6.7B (Wei et al., 2024) at 32.4%. While both these models perform slightly better, they used orders of magnitude more seed data; Magicoder used OSS-Instruct, requiring 80,000 code snippets compared to our 3 examples.

We note that with the 70B family on both HotpotQA and LCB, BARE-generated data leads to less of an improvement than the instruct-generated data. In these cases, using GPT-4o as a refiner instead of Llama-3.1-70B-Instruct does lead to stronger results, indicating that the choice of refiner can strongly influence data quality: the Llama model was unable to sufficiently refine the base model generations, but the stronger GPT-4o was able to do so. We show detailed results of refining with GPT-4o in all domains in Section B.2, emphasizing BARE's consistently strong data generation capabilities. For more detailed discussion, see Section C.

**Comparison to Real-World Data.** To further show the quality of our synthetic data, we took the real GSM8K training data (the only domain of these 4 which has a real train set) and randomly sampled 1,000 examples from which to fine-tune the Llama-3.2-1B-Instruct model. The resulting trained model had a performance of 26.6%, surpassing the 8B/70B Base and Instruct models individually. Interestingly, though, it was surpassed by almost all BARE methods, most prominently when GPT-4o was used as a refiner with Llama-3.1-70B-Base (35.8%). While one might expect the real training data should be better than a synthetically generated dataset based on a small sample, the quality of synthetically generated data with an emphasis on quality and diversity can lead to better generalization then the original human generated training data. This is in line with other works' findings for synthetic sets generated from large samples (Liu et al., 2023).

**Temperature, Scale, & Model Ablations.** We also find that temperature ablations do not meaningfully affect the utility of instruction-tuned model generations, especially when compared to gains using BARE; for details, see Section B.3. Additionally, we found that BARE's strong performance holds when scaling the amount of generated data; for details, see Section B.5. Finally, we found our results hold across model families such as the Qwen3 models; for details see Section B.6.

**Instruct-Instruct Ablation.** To verify BARE's performance comes from the use of base models rather than the multi-step setup, we replace the base model in the first step of BARE with an instruction-tuned model on the GSM8K task. Switching Llama-3.1-70B-Base to Instruct drops the accuracy on the test set from 29.8% to 25.4% when refining with Llama-3.1-70B-Instruct and from 35.8% to 30.8% with GPT-4o. At the same time, the average pairwise similarity before and after refinement remains similar, indicating that diversity must be introduced in the first stage. Having established that base models are most effective at seed efficient diversity, we conclude that the use of base models is a key reason for the utility of BARE. Detailed results from this ablation are in Section B.4.

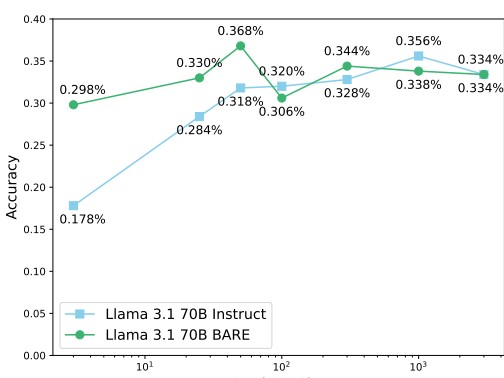

Figure 10: **Accuracy at varying seed (few-shot) sizes** of a Llama-3.2-1B-Instruct Model finetuned on synthetic data generated using Instruct and BARE methods on GSM8K. BARE performance is consistent throughout while Instruct improves with seed set size before converging with BARE.

**Seed Data Scale Ablation.** We additionally investigate the advantage of BARE compared to instruct-only methods as more seed data is made available (i.e., higher seed data collection effort). We evaluate results with BARE and Instruct using the Llama 3.1 70B family on GSM8K, ranging the seed set size from 3 to 3000 and continuing to generate 1,000 examples (Figure 10). Each generation randomly selects three examples from the seed set for the prompt.

BARE consistently outperforms Instruct generation for small seed sets, highlighting the potential of BARE in making high-quality synthetic data generation easier to adopt and realistic with minimal effort. The methods converge and plateau for large seed set sizes, indicating that diversity sourced from the base model's priors in BARE is always at least as effective as diversity from curated seed sets. In our experiments, convergence between BARE and Instruct generation occurred at around 100 examples, or 10% of the generated dataset size. We leave as future work the validation of the convergence point at larger dataset scales.

## 6 CONCLUSION

In this work we investigated the potential of using base models for few-shot synthetic data generation. Through this investigation, we find that base models are an effective source of diversity even when given minimal seed data, making them an attractive choice for data efficiency and low-effort synthetic data generation. These insights motivate the design of a system novelly leveraging base models in synthetic data generation, BARE.

Through extensive experiments, we validate the importance of each step in BARE and demonstrate its ability to preserve base model diversity while enhancing output quality. The use and analysis of base models specifically in BARE opens an avenue for much future work as almost no prior work to our knowledge has studied their innate value (before post-training).

Moreover, by fine-tuning on BARE-generated data for various domains, we underscore BARE's practical utility, consistently outperforming existing synthetic data generation methods on downstream tasks such as GSM8K and LiveCodeBench, in addition to RAFT, for which we set a new SOTA.

REPRODUCIBILITY STATEMENT

To support the reproducibility of our work, we will publicly release our code after the anonymization period. In the meantime, we provide a detailed description of our method in Section 4 and all prompts used in Appendix D. Detailed data generation parameters are provided in Appendix A and training parameters and evaluation setups are described in Section 5 and Appendix B.1.

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

## A  SAMPLING DETAILS

During data generation, all base models were sampled at a default temperature of 0.7. For instruction-tuned-only generations we sampled at the highest temperature at which we experimentally found data generation to still be coherent, which is 1.0 for Llama models and 1.2 for GPT-4o. However, for Enron, we sample from GPT-4o at a temperature of 1.0 to maintain generation coherence. Similarly, for Enron and Newsgroups, we sample for Llama-3.1-8B-Instruct at a temperature of 0.7. Instruction-tuned models used during refinement are always sampled at a temperauter of 0.7.

## B  ADDITIONAL RESULTS

### B.1  DOWNSTREAM EVALUATION - ADDITIONAL DETAILS

We list below the fine-tuning hyperparameters that were used in common for HotpotQA, PubMedQA, GSM8K, and LCB TOP. Learning rate was determined independently for each domain via learning rate sweeps (across orders of magnitude); each sweep gave the same optimal learning rate.

- Learning Rate: 0.001
- LoRA $\alpha$: 16
- LoRA Rank: 8
- LoRA Dropout: 0.0

The generated data is used to train a BERT-based classifier Devlin et al. (2018) for 2 epochs on Enron and 9 epochs on Newsgroups. The trained models are evaluated on a static test set with $n = 500$ examples for each domain.

We use 1-4 A100s for serving open-source models for data generation tasks (8B and 70B models) and up to a single node of 8xH100s for training purposes. LoRA fine-tuning takes on the order of minutes and data generation for 10,000 examples for an 70B model using 90 cores takes approximately an hour.

### B.2  CORE EXPERIMENT RESULTS - ALL DOMAINS

This appendix contains diversity, IR, and downstream performance results for all core experiments: generation with Llama 3.1 8B and 70B Base and Instruct models, BARE with Llama 3.1 models of both families, and BARE with the use of GPT-4o.

Note that HotpotQA RAFT and PubMedQA RAFT diversity results present here were not presented in Table 1 as we believe the numbers are noisy and not fit for drawing conclusions, due to the use of 100 different simulated retrieval contexts that generation was conditioned on (as required by RAFT). Not only does this introduce noise to the similarity calculation, but the strong instruction following capability of instruct models allow them to better leverage the inherent diversity in different prompts. However, for completeness, we report the values in the tables in this appendix. The fine-tuned models are evaluated on a static $n = 100$ test set for HotpotQA and PubMedQA, a static $n = 500$ test set for GSM8K, and the full $n = 442$ test set for LCB TOP.

Table 3: Average pairwise embedding cosine similarity, IR, and downstream F1 results on Enron. A BERT model with a classification head was trained for 2 epochs on the generated data. Only pairwise similarities for generations within the same class (spam or legitimate) were calculated.

| Generation Method | Average Embedding Similarity | IR | Downstream F1 |
|---|---|---|---|
| Llama 3.1 8B Instruct | 0.500 | 86.0% | 0.753 |
| Llama 3.1 70B Instruct | 0.450 | 85.0% | 0.848 |
| Llama 3.1 8B Base | 0.368 | 63.5% | 0.790 |
| Llama 3.1 70B Base | 0.350 | 74.5% | 0.819 |
| BARE Llama 3.1 8B | 0.413 | 85.0% | **0.872** |
| BARE Llama 3.1 70B | 0.406 | 82.0% | 0.771 |
| BARE GPT-4o + Llama 3.1 8B Base | 0.379 | 84.5% | **0.872** |
| BARE GPT-4o + Llama 3.1 70B Base | 0.356 | 88.5% | 0.846 |

Table 4: Average pairwise embedding cosine similarity, IR, and downstream accuracy results on Newsgroups. A BERT model with a classification head was trained for 9 epochs on the generated data.

| Generation Method | Average Embedding Similarity | IR | Downstream Accuracy |
|---|---|---|---|
| Llama 3.1 8B Instruct | 0.271 | 85% | 26% |
| Llama 3.1 70B Instruct | 0.246 | 82% | 30% |
| Llama 3.1 8B Base | 0.155 | 58% | 41% |
| Llama 3.1 70B Base | 0.162 | 78% | 29% |
| BARE Llama 3.1 8B | 0.162 | 91% | 40% |
| BARE Llama 3.1 70B | 0.134 | 93% | **49%** |
| BARE GPT-4o + Llama 3.1 8B Base | 0.131 | 81% | 44% |
| BARE GPT-4o + Llama 3.1 70B Base | 0.285 | 87% | 47% |

Table 5: Average pairwise embedding cosine similarity, IR, and downstream accuracy results on HotpotQA RAFT. A Llama-3.1-8B-Instruct model was fine-tuned for 4 epochs on the generated data. The baseline performance of the Llama-3.1-8B-Instruct model on the evaluation set prior to any fine-tuning is reported in the first row.

| Generation Method | Average Embedding Similarity | IR | Downstream Accuracy |
|---|---|---|---|
| Baseline Performance | – | – | 33% |
| Llama 3.1 8B Instruct | 0.214 | 76% | 49% |
| Llama 3.1 70B Instruct | 0.216 | 90% | 55% |
| Llama 3.1 8B Base | 0.221 | 62% | 50% |
| Llama 3.1 70B Base | 0.209 | 77% | 53% |
| BARE Llama 3.1 8B | 0.217 | 77% | **58%** |
| BARE Llama 3.1 70B | 0.210 | 88% | 54% |
| BARE GPT-4o + Llama 3.1 8B Base | 0.214 | 78% | 57% |
| BARE GPT-4o + Llama 3.1 70B Base | 0.205 | 89% | 56% |

Table 6: Average pairwise embedding cosine similarity, IR, and downstream accuracy results on PubMedQA RAFT. A Llama-3.1-8B-Instruct model was fine-tuned for 4 epochs on the generated data. The baseline performance of the Llama-3.1-8B-Instruct model on the evaluation set prior to any fine-tuning is reported in the first row.

| GENERATION METHOD | AVERAGE EMBEDDING SIMILARITY | IR | DOWNSTREAM ACCURACY |
|---|---|---|---|
| BASELINE PERFORMANCE | – | – | 53% |
| LLAMA 3.1 8B INSTRUCT | 0.376 | 63% | 72% |
| LLAMA 3.1 70B INSTRUCT | 0.373 | 73% | 73% |
| LLAMA 3.1 8B BASE | 0.396 | 39% | 68% |
| LLAMA 3.1 70B BASE | 0.603 | 31% | 73% |
| BARE LLAMA 3.1 8B | 0.377 | 47% | 73% |
| BARE LLAMA 3.1 70B | 0.367 | 72% | **79%** |
| BARE GPT-4O + LLAMA 3.1 8B BASE | 0.385 | 57% | 72% |
| BARE GPT-4O + LLAMA 3.1 70B BASE | 0.474 | 40% | 63% |

Table 7: Average pairwise embedding cosine similarity, IR, and downstream accuracy results on GSM8K. A Llama-3.2-1B-Instruct model was fine-tuned for 4 epochs on the generated data instead of the 3.1 8B model due to its high no-training performance. The baseline performance of the Llama-3.2-1B-Instruct model on the evaluation set prior to any fine-tuning is reported in the first row. Llama-3.1-8B-Instruct generation results are not reported due to data generation derailing.

| GENERATION METHOD | AVERAGE EMBEDDING SIMILARITY | IR | DOWNSTREAM ACCURACY |
|---|---|---|---|
| BASELINE PERFORMANCE | – | – | 21.8% |
| LLAMA 3.1 8B INSTRUCT | N/A | N/A | N/A |
| LLAMA 3.1 70B INSTRUCT | 0.421 | 51% | 17.8% |
| LLAMA 3.1 8B BASE | 0.310 | 23% | 19.2% |
| LLAMA 3.1 70B BASE | 0.313 | 44% | 22.4% |
| BARE LLAMA 3.1 8B | 0.310 | 27% | 24.8% |
| BARE LLAMA 3.1 70B | 0.305 | 54% | 29.8% |
| BARE GPT-4O + LLAMA 3.1 8B BASE | 0.295 | 60% | 32.8% |
| BARE GPT-4O + LLAMA 3.1 70B BASE | 0.302 | 64% | **35.8%** |

Table 8: Average pairwise embedding cosine similarity, IR, and downstream accuracy results on LCB TOP. A Llama-3.1-8B-Instruct model was fine-tuned for 4 epochs on the generated data. The baseline performance of the Llama-3.1-8B-Instruct model on the evaluation set prior to any fine-tuning is reported in the first row.

| GENERATION METHOD | AVERAGE EMBEDDING SIMILARITY | IR | DOWNSTREAM ACCURACY |
|---|---|---|---|
| BASELINE PERFORMANCE | – | – | 18.6% |
| LLAMA 3.1 8B INSTRUCT | 0.416 | 51% | 20.6% |
| LLAMA 3.1 70B INSTRUCT | 0.389 | 49% | 26.0% |
| LLAMA 3.1 8B BASE | 0.468 | 36% | 24.9% |
| LLAMA 3.1 70B BASE | 0.477 | 57% | 24.4% |
| BARE LLAMA 3.1 8B | 0.462 | 47% | **28.1%** |
| BARE LLAMA 3.1 70B | 0.481 | 64% | 25.6% |
| BARE GPT-4O + LLAMA 3.1 8B BASE | 0.459 | 72% | 26.7% |
| BARE GPT-4O + LLAMA 3.1 70B BASE | 0.471 | 68% | 27.4% |

## B.3 INDEPENDENT SAMPLING TEMPERATURE ABLATIONS - HOTPOTQA, PUBMEDQA, AND LCB TOP

This appendix contains diversity, IR, and downstream performance results for our temperature ablation experiments. We perform a temperature sweep for Llama-3.1-8B-Instruct generation with $t = 0.5, 0.7, 1.0$. We find that while adjusting the temperature can improve downstream performance, in general the gains are small relative to gains by using BARE.

Table 9: Temperature ablations with independent sampling from Llama-3.1-8B-Instruct. Average pairwise embedding cosine similarity, IR, and downstream accuracy results on HotpotQA RAFT. A Llama-3.1-8B-Instruct model was fine-tuned for 4 epochs on the generated data. The baseline performance of the Llama-3.1-8B-Instruct model on the evaluation set prior to any fine-tuning is reported in the first row.

| GENERATION METHOD | AVERAGE EMBEDDING SIMILARITY | IR | DOWNSTREAM ACCURACY |
|---|---|---|---|
| BASELINE PERFORMANCE | – | – | 33% |
| LLAMA 3.1 8B INSTRUCT ($t = 1.0$) | 0.214 | 76% | 49% |
| LLAMA 3.1 8B INSTRUCT ($t = 0.7$) | 0.216 | 77% | 50% |
| LLAMA 3.1 8B INSTRUCT ($t = 0.5$) | 0.220 | 83% | 42% |
| LLAMA 3.1 8B BASE ($t = 1.0$) | 0.221 | 62% | 50% |
| BARE LLAMA 3.1 8B (ALL $t = 0.7$) | 0.217 | 77% | **58%** |

Table 10: Temperature ablations with independent sampling from Llama-3.1-8B-Instruct. Average pairwise embedding cosine similarity, IR, and downstream accuracy results on PubMedQA RAFT. A Llama-3.1-8B-Instruct model was fine-tuned for 4 epochs on the generated data. The baseline performance of the Llama-3.1-8B-Instruct model on the evaluation set prior to any fine-tuning is reported in the first row.

| GENERATION METHOD | AVERAGE EMBEDDING SIMILARITY | IR | DOWNSTREAM ACCURACY |
|---|---|---|---|
| BASELINE PERFORMANCE | – | – | 53% |
| LLAMA 3.1 8B INSTRUCT ($t = 1.0$) | 0.376 | 62% | 72% |
| LLAMA 3.1 8B INSTRUCT ($t = 0.7$) | 0.375 | 71% | 72% |
| LLAMA 3.1 8B INSTRUCT ($t = 0.5$) | 0.377 | 73% | **75%** |
| LLAMA 3.1 8B BASE ($t = 1.0$) | 0.396 | 39% | 68% |
| BARE LLAMA 3.1 8B (ALL $t = 0.7$) | 0.377 | 47% | 73% |

Table 11: Temperature ablations with independent sampling from Llama-3.1-8B-Instruct. Average pairwise embedding cosine similarity, IR, and downstream accuracy results on the full LCB TOP set. A Llama-3.1-8B-Instruct model was fine-tuned for 4 epochs on the generated data. The baseline performance of the Llama-3.1-8B-Instruct model on the evaluation set prior to any fine-tuning is reported in the first row.

| GENERATION METHOD | AVERAGE EMBEDDING SIMILARITY | IR | DOWNSTREAM ACCURACY |
|---|---|---|---|
| BASELINE PERFORMANCE | – | – | 18.6% |
| LLAMA 3.1 8B INSTRUCT ($t = 1.0$) | 0.365 | 33% | 20.1% |
| LLAMA 3.1 8B INSTRUCT ($t = 0.7$) | 0.416 | 51% | 20.6% |
| LLAMA 3.1 8B INSTRUCT ($t = 0.5$) | 0.450 | 53% | 22.9% |
| LLAMA 3.1 8B BASE ($t = 1.0$) | 0.468 | 36% | 24.9% |
| BARE LLAMA 3.1 8B (ALL $t = 0.7$) | 0.462 | 47% | **28.1%** |

## B.4 BARE FIRST STAGE ABLATIONS - GSM8K

This appendix contains diversity, IR, and downstream performance results for our ablation replacing the first stage of BARE with an instruction-tuned model, specifically Llama-3.1-70B-Instruct. We refine using Llama-3.1-70B-Instruct and GPT-4o, and investigate the change in downstream performance compared to standard BARE (using Llama-3.1-70B-Base in the first stage). Note that dataset diversity is unchanged compared to direct generation from Llama-3.1-70B-Instruct, that IR improves after refinement, and that downstream performance is consistently worse than standard BARE.

Table 12: Average pairwise embedding cosine similarity, IR, and downstream accuracy results on GSM8K. A Llama-3.2-1B-Instruct model was fine-tuned for 4 epochs on the generated data. The baseline performance of the Llama-3.2-1B-Instruct model on the evaluation set prior to any fine-tuning is reported in the first row.

| GENERATION METHOD | AVERAGE EMBEDDING SIMILARITY | IR | DOWNSTREAM ACCURACY |
|---|---|---|---|
| BASELINE PERFORMANCE | – | – | 21.8% |
| LLAMA 3.1 70B INSTRUCT | 0.421 | 51% | 22.4% |
| LLAMA 3.1 70B INSTRUCT SELF-REFINE | 0.422 | 63% | 25.4% |
| GPT-4O REFINING LLAMA 3.1 70B INSTRUCT | 0.421 | 70% | 30.8% |
| BARE LLAMA 3.1 70B | 0.305 | 54% | 29.8% |
| BARE GPT-4O + LLAMA 3.1 70B BASE | 0.302 | 64% | **35.8%** |

## B.5 BARE TRAINING SET SCALE ABLATIONS

This appendix contains downstream performance results for our ablation where we scale up the number of BARE generated data points from $1,000$ to $10,000$ in an effort to investigate whether the trends hold at larger scales. We focused our investigation of this particular ablation on the Llama-3.1-8B model family, and across both the LCB TOP and PubMedQA domains we find that the BARE trends consistently hold at scale.

Table 13: Downstream accuracy results on two evaluation sets (LCB and PubMedQA) using 10,000 synthetically-generated data points and the Llama-3.1-8B model family. Models were fine-tuned for 2 epochs rather than the 4 used for 1,000 points due to compute considerations.

| EVALUATION SET | BASE | INSTRUCT | BARE |
|---|---|---|---|
| LCB TOP | 0.192 | 0.181 | **0.233** |
| PUBMEDQA | 0.512 | 0.645 | **0.658** |

## B.6 BARE MODEL FAMILY ABLATIONS

This appendix presents downstream performance results for our ablation where we use the Qwen3 (Yang et al., 2025) model family for data generation to verify that BARE works across model families. Specifically, we evaluated on the 8B model sizes on the GSM8K domain. Here, the greater diversity from base model generations continues to lead to improved performance over Instruct, indicating the utility of BARE generalizes to other model families.

Table 14: Downstream accuracy results on GSM8K using 1,000 synthetically-generated data points and the Qwen3-8B model family. All training/sampling parameters were the same as for the Llama-3.1 experiments.

| | BASE | INSTRUCT | BARE |
|---|---|---|---|
| ACCURACY | 0.312 | 0.382 | **0.392** |

## C ADDITIONAL DISCUSSION

In this appendix we further discuss our evaluation results and propose hypotheses for certain trends or observations.

**Why is BARE generation sometimes worse than instruction-tuned generation?** As mentioned in the main paper, the choice of refiner plays an important role in the BARE pipeline. A model that is proficient at generating data may not be proficient at improving data. Mitigation of this issue can include tailoring prompts for specific models, but we chose to keep our prompts consistent and general to focus on the core idea of leveraging base mode generations.

**Why is instruction-tuned generation sometimes worse than base generation?** A key insight to our work is the diversity-quality tradeoff between base models and instruction-tuned models. In some cases, the additional diversity from base models outweighs the lower quality.

**Why do smaller models sometimes produce better data than larger models?** The diversity-quality tradeoff between smaller and larger models is not well understood. While it is generally accepted that larger models are of higher quality, it is unclear whether they are similar in terms of diversity. In some cases, the smaller model exhibits higher diversity in its generations, allowing the resulting datasets to be of more utility despite the lower quality.

## D  GSM8K PROMPT EXAMPLES

In this appendix, we provide exact prompts used for the GSM8K domains, representative of those used throughout this work. Examples are formatted for inclusion in the prompts in the "{examples}" fields, with "EXAMPLE START" and "EXAMPLE END" delimiters for the base prompt. BARE uses the standard Base Prompt in the base generation step.

STATIC FEW-SHOT EXAMPLES

---

**Example 1**

**Question:** Alice has 20 quarters. She wants to exchange them for nickels and so she goes to the bank. After getting back from the bank, she discovers that 20% of the nickels are iron nickels worth \$3 each. What is the total value of her money now?
**Answer:** A quarter is worth five nickels because .25 / .05 = $\ll .25/.05 = 5 \gg$ 5. She gets 100 nickels from the bank because 20 x 5 = $\ll 20 * 5 = 100 \gg$ 100. 20 of the nickels are iron nickels because 100 x .20 = $\ll 100 * .20 = 20 \gg$ 20. 80 of the nickels are regular because 100 - 20 = $\ll 100 - 20 = 80 \gg$ 80. The iron nickels are worth \$60 because 20 x 3 = $\ll 20 * 3 = 60 \gg$ 60. The regular nickels are worth \$4 because 80 x .05 = $\ll 80 * .05 = 4 \gg$ 4. Her money is now worth \$64 because 60 + 4 = $\ll 60 + 4 = 64 \gg$ 64. #### 64

---

**Example 2**

**Question:** A church has 120 members. 40% are adults. The rest are children. How many children more children are there than adults?
**Answer:** There are 48 adults because 120 x .4 = $\ll 120 * .4 = 48 \gg$ 48. 60% of members are children because 100 - 40 = $\ll 100 - 40 = 60 \gg$ 60. There are 72 children because 120 x .6 = $\ll 120 * .6 = 72 \gg$ 72. There are 24 more children than adults because 72 - 48 = $\ll 72 - 48 = 24 \gg$ 24. #### 24

---

**Example 3**

**Question:** Lisa is looking to attempt a World Record. She has decided to try and match Joey Chestnut's record of eating 75 full hotdogs, buns included, in 10 minutes. Halfway through the time Lisa has eaten 20 hotdogs. How many hotdogs will she have to eat per minute to at least tie Joey Chestnut's record?
**Answer:** Joey Chestnut ate 75 hotdogs to claim the record and Lisa has eaten 20 hot dogs so far, so she still needs to eat $75 - 20 = \ll 75 - 20 = 55 \gg$ 55 hotdogs to tie Joey Chestnut. Lisa has a 10-minute time period to eat the hotdogs and half the time has already passed, which means Lisa has $10/2 = \ll 10/2 = 5 \gg$ 5 minutes left until the competition is over. If she needs to eat 55 hotdogs to tie Joey Chestnut and there are 5 minutes left in the competition period, then she needs to eat $55/5 = \ll 55/5 = 11 \gg$ 11 hot dogs per minute to have a chance of tying for a win. #### 11

**Base Prompt**

Here are a few examples of grade school math word problems that require performing a sequence of elementary calculations using basic arithmetic operations. A bright middle school student should be able to solve each problem. The numerical answer is provided at the end of each example after ####.

{examples}

EXAMPLE START

**Instruct Few-shot Prompt**

Provide an example of a grade school math word problem that requires performing a sequence of elementary calculations using basic arithmetic operations. A bright middle school student should be able to solve each problem. Problems require no concepts beyond the level of early Algebra. You must first specify the question, then provide the very concise reasoning and answer. Provide your example in the following format:

```
Question:  [question]
Answer:  [answer]
```

Provide only the question and answer in the given format. Note how the numerical answer is provided after #### after each brief reasoning for a question. Here are some examples:

{examples}

Now it's your turn. Start your response with the question.

**Refine Prompt**

Improve the given grade school math word problem. Edit the problem or answer to be more similar in style to the examples, and disambiguate as necessary, in addition to correcting any errors. Do not change the theme of the problem. A bright middle school student should be able to solve each problem. Problems require no concepts beyond the level of early Algebra. Note how the numerical answer is provided after #### after each brief reasoning for a question. Provide your edited problem in the following format:

```
Question:  [question]
Answer:  [answer]
```

Provide only the question and answer in the given format. Here are some examples of categories and problems on those categories:

{examples}

Now it's your turn. Here is the question and answer for you to edit:
Question:
{question}
Answer:
{answer}

Provide only the improved question and answer in the given format. Do not include any commentary or notes. Start your response with the question.

**Sequential Prompt**

Generate a new grade school math word problem that requires performing a sequence of elementary calculations using basic arithmetic operations. A bright middle school student should be able to solve each problem. Problems require no concepts beyond the level of early Algebra. Here are the previously generated examples:

{examples}

Your new problem should:

     1.  Be different from the previous examples

     2.  Follow the same format and style as prior problems

Note how the numerical answer is provided after #### after each brief reasoning for a question. Provide only the question and answer in the given format here:

```
Question:  [question]
Answer:  [answer]
```

Start your response with the question.

**In One Prompt**

Provide {num} examples of problems that might be grade school math word problems that require performing a sequence of elementary calculations using basic arithmetic operations. A bright middle school student should be able to solve each problem. Problems require no concepts beyond the level of early Algebra. You must first specify the question then provide the brief reasoning and answer. Note how the numerical answer is provided after #### after each brief reasoning for a question. Provide your examples in the following format:

```
Question:  [question]
Answer:  [answer]
```

Here are some examples:

{examples}

Now it's your turn. Generate {num} different problems following this format. Your question should be different in content from the examples. Make sure to only provide only the question and answer. Start each example with the question. Delimit the end of an example with the phrase "END OF EXAMPLE" (all caps) on a new line.

---

**Persona Prompt**

{persona_description}

Provide an example of a grade school math word problem that requires performing a sequence of elementary calculations using basic arithmetic operations. A bright middle school student should be able to solve each problem. Problems require no concepts beyond the level of early Algebra. You must first specify the question, then provide the very concise reasoning and answer. Provide your example in the following format:

```
Question:  [question]
Answer:  [answer]
```

Provide only the question and answer in the given format. Note how the numerical answer is provided after #### after each brief reasoning for a question. Here are some examples:

{examples}

Now it's your turn. Start your response with the question.

---

**Indistinguishability Rate Prompt**

System Prompt:

You are an expert at evaluating question and answer pairs for grade school math word problems.

You will be shown {k} examples. Each example consists of some context, a question, and an answer. All but one of them is generated from a high quality AI while one of them is of low quality.

Your task is to identify which example (1, 2, ..., {k}) appears to be of low quality. Consider factors like:

1. Differing natural language patterns and flow
2. Differing question structure, clarity, and difficulty
3. Context and specificity
4. Any subtle artifacts or unnatural patterns

Analyze each example carefully and explain your reasoning. End with 'Answer: [Question Number]' where Question Number is 1, 2, ..., {k}.

—

User Prompt:

Here are {k} examples. One of them is of low quality. Please identify which one:

{questions}

Analyze each example and explain which one you think is of low quality. End with 'Answer: [Question Number]'.

---

# E  USE OF LLMS

In this work, we used LLMs as a general-purpose assistant tool. LLMs assisted with editing tasks, such as making writing more concise and clear and formatting tables figures. All text included in the paper was originally written by an author and went through a final pass from an author.

We additionally used LLMs as a coding assistant. LLMs assisted with implementation and figure creation.

