# OpenReview forum: "BARE: Leveraging Base Language Models for Few-Shot Synthetic Data Generation"
_ICLR.cc/2026/Conference — Submitted to ICLR 2026_

### Official Review · Reviewer_gmgL · 2025-10-30

**Soundness:** 2
**Presentation:** 3
**Contribution:** 2
**Rating:** 4
**Confidence:** 3

**Summary:**

This paper introduces Base-Refine (BARE), a novel two-stage method that tackles the quality-diversity trade-off in few-shot synthetic data generation. BARE first leverages a base model for diverse data generation from minimal seed examples, then uses a capable instruction-tuned model to individually refine these outputs for high quality, correcting errors while preserving diversity. The authors substantiate the framework's efficacy with experiments across a wide range of domains.

**Strengths:**

1.	The manuscript is well-written, the proposed BARE pipeline is well-presented and can be easily understood.

2.	The authors validate their BARE framework across a wide range of domains, from natural language classification to complex reasoning in math and code.

**Weaknesses:**

1.	Limited Novelty: The conceptual novelty of the proposed BARE framework appears incremental. The "refine" stage, a core component, seems to be a direct application of existing self-correction techniques [1] to the domain of LLM training data. The authors should more clearly articulate the fundamental innovation of this stage beyond a simple application of prior work.

2.	Insufficient Motivation: The paper's central motivation, tackling the quality-diversity trade-off in few-shot synthetic data generation, is not well-supported. This premise is challenged by recent works like Magpie [2], which can generate high-quality and diverse data from scratch (i.e., in a zero-shot setting) without requiring seed examples. The authors must provide a clearer justification for why such methods are insufficient and why the few-shot problem setting remains a critical, unsolved challenge.

3.	Insufficient Experimental Evaluation: The empirical evaluation is lacking crucial comparisons to state-of-the-art baselines. To properly contextualize the performance of BARE, the experiments should include more methods from the post-training data synthesis literature, such as [2-6]. Without these comparisons, the claimed advantages of the BARE framework are not adequately substantiated.



[1] Kamoi, Ryo, et al. "When can llms actually correct their own mistakes? a critical survey of self-correction of llms." Transactions of the Association for Computational Linguistics 12 (2024): 1417-1440.

[2] Xu, Zhangchen, et al. "Magpie: Alignment Data Synthesis from Scratch by Prompting Aligned LLMs with Nothing." The Thirteenth International Conference on Learning Representations, 2025.

[3] Xu, Can, et al. "WizardLM: Empowering large pre-trained language models to follow complex instructions." The Twelfth International Conference on Learning Representations. 2024.

[4] Ding, Ning, et al. "Enhancing Chat Language Models by Scaling High-quality Instructional Conversations." Proceedings of the 2023 Conference on Empirical Methods in Natural Language Processing. 2023.

[5] Lambert, Nathan, et al. "T" ulu 3: Pushing frontiers in open language model post-training." arXiv preprint arXiv:2411.15124 (2024).

[6] Kaur, Simran, et al. "Instruct-skillmix: A powerful pipeline for llm instruction tuning." arXiv preprint arXiv:2408.14774 (2024).

**Questions:**

1.	How does the "refine" stage differ conceptually from a direct application of the self-correction techniques outlined in [1]?

2.	Given that methods like Magpie [2] can generate high-quality, diverse data from scratch, why is the quality-diversity trade-off in few-shot generation still a key motivation?

3.	Can the authors provide theoretical discussion and empirical results to show that the proposed method can outperform the previous data synthesis methods like [2-6]?

[1] Kamoi, Ryo, et al. "When can llms actually correct their own mistakes? a critical survey of self-correction of llms." Transactions of the Association for Computational Linguistics 12 (2024): 1417-1440.

[2] Xu, Zhangchen, et al. "Magpie: Alignment Data Synthesis from Scratch by Prompting Aligned LLMs with Nothing." The Thirteenth International Conference on Learning Representations, 2025.

[3] Xu, Can, et al. "WizardLM: Empowering large pre-trained language models to follow complex instructions." The Twelfth International Conference on Learning Representations. 2024.

[4] Ding, Ning, et al. "Enhancing Chat Language Models by Scaling High-quality Instructional Conversations." Proceedings of the 2023 Conference on Empirical Methods in Natural Language Processing. 2023.

[5] Lambert, Nathan, et al. "T" ulu 3: Pushing frontiers in open language model post-training." arXiv preprint arXiv:2411.15124 (2024).

[6] Kaur, Simran, et al. "Instruct-skillmix: A powerful pipeline for llm instruction tuning." arXiv preprint arXiv:2408.14774 (2024).

---

> ### Author Response · Authors · 2025-11-19
>
> We thank the reviewer for their constructive evaluation, especially the recognition of the importance of the problem setting and the depth of our empirical analyses. Below we address the concerns in detail.
>
> ### Novelty
>
> We agree that the refinement step itself is not algorithmically novel; we do not claim novelty in refinement in the paper. Our contribution comes from the empirical finding that:
>
> - Base models exhibit substantially higher semantic diversity than instruction-tuned models in the few-shot regime,
> - Instruct models systematically collapse this diversity, and
> - Cross-model refinement (instruct refiner over base drafts) produces data that is both diverse and high-quality, whereas neither base-only nor instruct-only generation achieves this balance.
>
> Self-correction work, such as [1], assume the same model can recognize and repair its own errors, a condition that does not hold for base models which cannot reliably follow instructions or maintain structure. BARE introduces the idea that base models should serve as the drafter, precisely because instruct models collapse diversity, and instruct models should serve as the refiner, precisely because base models cannot self-correct.
>
> ### Few-Shot Motivation
>
> Magpie-style zero-shot synthesis addresses a different problem: producing general alignment/instruction-following interactions. It does not attempt to replicate domain-specific task behavior, formats, or conventions. In real applications, datasets often encode task-specific structure, custom schemas, or implicit labeling conventions. Zero-shot prompting cannot reproduce these reliably and base models cannot be guided without seed examples, as they do not reliably follow instructions.
>
> Few-shot synthetic data generation is therefore essential whenever the goal is to recreate or extend task-specific datasets. Magpie does not solve for this setting; it solves a different alignment-oriented synthesis problem. We will make this distinction clearer in an updated version of the related works.
>
> ### Baselines
>
> We appreciate the request for broader comparisons, but these methods are not designed for few-shot, task-specific data synthesis, and thus do not serve as fair baselines as we discuss in the related works section.
>
> These methods operate in a fundamentally different regime from ours. They rely on large seed sets, or multi-stage post-training corpora, or general instruction-following conversational data.
>
> In contrast, BARE is explicitly designed for minimal-seed synthetic data generation, utilizing just 3 few-shot examples per task. This setting is motivated by realistic low-data industry workflows, where acquiring even dozens of examples may be expensive.
>
> WizardLM uses a 52K seed instruction set to evolve its dataset, Magpie focuses only on general alignment (it cannot take in a description of a task), Instruct-SkillMix seeds with 500 different skills, and lastly Tulu3 uses the persona-prompting approach which we do baseline against in the paper.
>
> Given constant output dataset size, the other fair baselines would include instruction-only generation (repeated sampling from an instruction-tuned model), and instruct-instruct pipelines (refinement without base models). We also baseline against various prompting methods. These match the few-shot constraints while preserving the dataset size and generation capacity. These are precisely the baselines we evaluate, and BARE significantly outperforms them.
> We will clarify this distinction further and expand the discussion of seed requirements.

---

### Official Review · Reviewer_mjgC · 2025-10-31

**Soundness:** 2
**Presentation:** 3
**Contribution:** 2
**Rating:** 4
**Confidence:** 3

**Summary:**

This paper introduces BARE (Base-Refine), a two-stage framework for few-shot synthetic data generation. The authors observe that instruction-tuned LLMs produce high-quality but low-diversity data, while base LLMs generate diverse but lower-quality samples. BARE leverages this complementarity:
(i) A base model first generates a diverse pool of synthetic data from only a few seed examples (e.g., 3).
(ii) An instruction-tuned model then refines each generated example for correctness and clarity.

The paper demonstrates that BARE produces diverse and realistic datasets that significantly improve downstream fine-tuning performance, even with minimal seed data. For instance, fine-tuning Llama-3.2-1B on 1,000 BARE-generated GSM8K samples improves accuracy by 101% over instruction-only baselines, and surpasses strong methods such as RAFT on RAG tasks by 18.4%.

**Strengths:**

1. The observation that base models retain diversity while instruct models lose it due to post-training is well-motivated and empirically verified.
2. BARE is a simple yet effective way to combine strengths of base and instruct models, addressing the overlooked potential of base models in data generation.

**Weaknesses:**

1. While combining base and instruct models is insightful, the method itself (generate + refine) is conceptually simple and resembles earlier “draft–refine” pipelines. The main novelty lies in the empirical insight rather than algorithmic design.
2. The success of BARE heavily depends on the choice and strength of the refiner model (e.g., GPT-4o vs. Llama-Instruct), which limits reproducibility for weaker setups.

**Questions:**

1. How sensitive is BARE’s performance to the choice of refiner model? For example, if a weaker instruction-tuned model (e.g., Llama-3-Instruct-8B) is used instead of GPT-4o, how much does performance degrade?
2. During refinement, how does the refiner avoid “mode collapse” and preserve diversity? Would it make more sense to impose some mechanism (e.g., randomness, sampling temperature) to prevent homogenization during refinement?
3. It would also be great to place larger / more recent LLMs as baselines and validate the proposed method.

---

> ### Author Response · Authors · 2025-11-19
>
> We thank the reviewer for their constructive evaluation, especially the recognition of the depth of our empirical analysis and the simplicity of BARE’s leveraging of base models. Below we address the concerns in detail.
>
> ### Novelty
>
> We appreciate the observation and agree that “generate + refine’’ is a known structural pattern. However, **the core novelty of BARE is not the existence of two stages but which models should occupy these stages and why**. Prior pipelines overwhelmingly use *instruction-tuned* models for both steps, or use them as drafters because of their stability. In contrast, our work:
>
> 1. Identifies and quantifies an underappreciated property:
> base models retain significantly higher semantic diversity even in the few-shot regime, whereas instruct models collapse early.
>
> 2. Establishes that this diversity is crucial for few-shot synthetic data generation, and that instruction models cannot recover it even with advanced prompting (Table 2), seed variation, or higher temperatures.
>
> 3. Demonstrates that base-generated diversity can be made usable through refinement—an empirical question that was far from obvious, given the derailment patterns of base models.
>
> We believe this empirical insight is itself a meaningful contribution: it changes the assignment of roles within data-generation systems, and it leads to large, consistent downstream gains across tasks. Indeed, the reviewer themself notes the novelty of this insight as a strength of the paper. Many impactful papers introduce simple pipelines whose significance comes from *the choice of components and the evidence supporting that choice.*
>
> ### Impact of Refiner Strength
>
> Our results show that BARE remains effective even with substantially weaker refiners:
>
> - In Tables across Sections 5 and B.2, Llama-3.1-8B-Instruct, a comparatively weak model, consistently improves base generations and yields strong downstream accuracy.
> - GPT-4o naturally provides higher ceilings, but the qualitative behavior of BARE (i.e., the preservation of diversity and improvement in entry-wise quality) holds across all examined refiners.
>
> We view the refiner as a task-dependent hyperparameter, just as prior data-generation pipelines tune sampling temperature, filtering heuristics, or prompt structure. Importantly, BARE improves over instruct-only pipelines even when both the generator and refiner are within the same family and scale (e.g., 8B or 70B), demonstrating that the method is not reliant on a single powerful model. Sensitivity results with other refiners are included in all tables/graphs in the paper.
>
> ### Diversity
>
> Diversity is preserved because the refiner is strongly conditioned on the base-model draft, and the refinement instruction explicitly requires maintaining the original concept and structure. Empirically: Figure 6 shows that the pairwise embedding distributions of BARE outputs are nearly identical to those of the base model drafts. Thus, refinement consistently improves entry-wise quality (Figure 7) without collapsing diversity, and no additional randomness is required.
>
> ### Larger Models
>
> Extending fine-tuning beyond the 8B scale was beyond our compute budgets, but our numerous results and ablations already validate that BARE’s benefits are not tied to a particular model size and scale with dataset size.

---

### Official Review · Reviewer_Qc13 · 2025-11-01

**Soundness:** 3
**Presentation:** 3
**Contribution:** 2
**Rating:** 4
**Confidence:** 3

**Summary:**

This paper proposes BARE, which is a new method for few-shot training data generation with base LLMs. Specifically, instead of synthesizing data with instruct model or base model only, BARE first generates training data with base model and then uses instruct model to further improve these data. In this way, the diversity of the synthetic data generated by the base model is maintained while the quality is improved by the instruct model. Evaluation shows that BARE-generated data performs the best for LoRA fine-tuning.

**Strengths:**

- This paper explores an important and interesting research direction.
- The evaluation has included lots of in-depth study and discussion on the quality of synthetic data.

**Weaknesses:**

- One major concern is the scalability of the method. While the evaluation has included different model series (i.e., Llama-3.1 and Qwen3) and benchmarks, the models used for training is limited to 8B-level and the training set only includes 1000 samples. More experiments on larger models with much larger training sets are needed to study whether the performance gain BARE brings can scale up consistently.
- The relationship between downstream accuracy and indistinguishability rate is unclear. Specifically, while the indistinguishability rate of BARE data is similar to that of instruct data, the training performance using BARE data is clearly better than that using instruct data. More study is needed here to show whether indistinguishability rate is a good metric to evaluate the quality of synthetic data.

**Questions:**

- Why does the evaluation focus on LoRA fine-tuning instead of full-weight fine-tuning?
- Results in Table 8 seem to show that “BARE LLAMA 3.1 70B” performs worse than “BARE LLAMA 3.1 8B” on downstream accuracy. What’s the explanation for this abnormal phenomenon?

---

> ### Author Response · Authors · 2025-11-19
>
> We thank the reviewer for their constructive evaluation, especially the recognition of the importance of the problem setting and the depth of our empirical analyses. Below we address the concerns in detail.
>
> ### Scalability of BARE
>
> Our paper already includes substantially larger-scale experiments, up to 10,000 generated samples, as reported in Appendix B.5. These results show that the benefits of BARE scale consistently with dataset size, and the gains over instruct-only generation remain robust. We apologize if this was unclear in the main text; we will add clearer references to these results.
>
> Regarding model size: the purpose of BARE is to improve synthetic data quality in few-shot settings, independent of the fine-tuning model’s scale. Regardless, we do include experiments using 70B-scale generators and refiners for data generation, and we additionally demonstrate that BARE’s advantages persist across model families (Appendix B.6, Qwen3). Extending fine-tuning at 70B scale was beyond our compute budgets, but our results already validate that BARE’s benefits are not tied to a particular model size and scale with dataset size.
>
> ### IR and Accuracy
>
> We agree with the reviewer that downstream accuracy is the ultimate measure of synthetic data quality, and we do not propose IR as a substitute. IR is explicitly introduced as a sample-level heuristic to understand realism before training, not as a predictor of task accuracy.
>
> The key point of the analysis is that IR helps reveal how base models differ from instruction-tuned models in entry-wise quality, while diversity metrics capture dataset-level properties, thus motivating the eventual method. BARE’s advantage arises because it preserves base-model diversity while correcting sample-level issues during refinement. The fact that IR is similar while downstream accuracy improves highlights that dataset-level diversity is essential, and IR alone cannot capture this. We will clarify this distinction.
>
> We appreciate the reviewer’s suggestion, and exploring improved heuristics for synthetic data quality is an important direction for future work.
>
> ### Why LoRA fine-tuning?
>
> LoRA fine-tuning is standard practice in many papers due to compute constraints. Prior work consistently shows that LoRA and full-weight fine-tuning provide highly correlated improvements, especially for classification and moderate-length generation tasks [1, 2].
>
> ### Why Does BARE (Llama-3.1-70B) Underperform BARE (Llama-3.1-8B) in Table 8?
>
> We hypothesize for reasons why this behavior might occur in Appendix C.3. One potential reason is the larger model’s initial generations have lower diversity (mode concentration at high capabilities) compared to the smaller model. The consequence of this intuition is further supported by our ablations in Appendix B.4, which show that diversity must originate in the first stage, and overly strong base models can occasionally reduce diversity.
>
> ### References
>
> [1] Schulman and Thinking Machines Lab (2025). LoRA Without Regret. https://thinkingmachines.ai/blog/lora/
>
> [2] Biderman, et al (2024). LoRA Learns Less and Forgets Less. https://arxiv.org/abs/2405.09673

---

### Official Review · Reviewer_C12M · 2025-11-01

**Soundness:** 2
**Presentation:** 2
**Contribution:** 1
**Rating:** 2
**Confidence:** 4

**Summary:**

Synthetic datasets are used to train LLMs, but they rely on a large number of curated seed tasks to introduce diversity into the synthetic data. This paper proposes BARE, a two-step approach that uses a base LM to generate highly diverse synthetic data and then an instruction-tuned model to improve generation quality. Furthermore, the base model uses only three seed examples, making it particularly useful for few-shot synthetic data generation. The downstream evaluations show that BARE-generated instructions improve LLM performance by a significant margin compared to baselines.

**Strengths:**

The paper is well motivated and well written.

The proposed approach is quite simple and easy to implement.

The downstream evaluation with synthetic data shows positive results compared to the baselines. In particular, the results on the LCB leaderboard with BARE instructions are quite promising.

**Weaknesses:**

**Missing key related work.**
Many of the findings and techniques in this paper have been introduced in prior work [a, b]. However, the paper does not discuss any of these works, either in the related work or the experiments. URIAL [a] uses three prompts to enable instruction-following abilities in base models. BARE also uses a similar three-prompt strategy to generate synthetic data with the base model. The authors should highlight the differences between the two methods. Furthermore, ALMA [b] uses only a base model for synthetic data generation, rather than the two-step approach with an instruction-tuned model in BARE. The paper should discuss and compare these works in the related work and experiments. Finally, the authors should also discuss related work that highlights the limitations of relying solely on the base model with in-context learning for following and generating instructions [c].

**Method.**
The method does not rely solely on the base model; BARE uses the base model to generate low-quality synthetic data and then an instruction-tuned model to refine it. While this is a reasonable approach, it still relies heavily on the instruction-tuned model to refine instructions. Furthermore, using the base model to produce more diverse synthetic data and then using an instruction-tuned model to improve the quality seems like a workaround rather than a principled approach to the problem. This could limit the impact of the work.

**Factual inaccuracies.**
In lines 89–90, the authors claim that this is the first work to show the value of base models for data generation. This is not accurate because [b] has already shown the benefits of using a base model for synthetic data generation. It would be great if the authors could refine or qualify this claim.

In line 42, the authors suggest that Li et al., 2024c [d] uses 500,000 text segments as seed sets. This is not accurate and is potentially misleading. Table 1 in [d] says they use 3,200 samples. Please consider rephrasing the sentence.

**Miscellaneous**
Figure 8: Missing Instruct result for GSM8K. The caption says “generations derailed.” What does this mean?

Entry-wise quality: It is unclear if this is a reliable metric. This appears to be an LLM-as-a-judge heuristic to indicate whether an example is synthetic or not. I’m not sure whether LLMs—even powerful ones like GPT-4o—can reliably judge whether an example is synthetic.

**References**

[a] *The Unlocking Spell on Base LLMs: Rethinking Alignment via In-Context Learning.* ICLR 2024.

[b] *ALMA: Alignment with Minimal Annotation.* arXiv 2024.

[c] *Is In-Context Learning Sufficient for Instruction Following in LLMs?* ICLR 2025.

[d] *Self-Alignment with Instruction Backtranslation.* ICLR 2024.

**Questions:**

See weaknesses.

---

> ### Author Response · Authors · 2025-11-19
>
> We would like to begin by thanking the reviewer for their constructive feedback. We appreciate the recognition of our strong empirical results, clarity, and the potential value of BARE for reducing the reliance on large curated seed datasets. Below, we clarify several misunderstandings, address suggestions for additional analysis, and provide details on experimental design choices.
>
> ### Related Works
>
> Thank you for pointing us to these works. We believe that there are several differences between these works that in some cases don’t make them related to our paper. One common key difference between those works and BARE is that BARE focuses on reasoning tasks while the works focus on general instruction-following alignment, a notably different setting.
>
> URIAL [a] improves base model alignment through few-shot in-context demonstrations, but few-shot examples tend not to significantly improve model performance on reasoning tasks. “Is In-Context Learning Sufficient For Instruction Following in LLMs” [c] challenges the results from [a] when compared against fine-tuning, again focusing on the few-shot in-context instruction-following problem.
>
> ALMA [b] is the most related work to ours. However, they focus on training base models for instruction following and require 9000 labeled examples, whereas BARE trains for reasoning tasks and requires just 3 examples.
>
> Critically, all three works investigate methods to improve the instruction-following ability of base models. Our setting is entirely different, where we aim to improve the reasoning capabilities of instruction-tuned models. While we do use base models in our pipeline, we do not rely on its instruction-following capabilities. Empirically, we found that for our reasoning tasks, base models with few-shot prompting did well enough at generating problems following the desired format, and when they did fail, refinement with the instruction-tuned model was able to fix the errors. BARE does not rely on the instruction-following capabilities of the base model; we only leverage the diversity of base models, and leverage the instruction-following capabilities of instruction-turned models to generate high-quality training data.
>
> ### BARE Uses Base and Instruction-Tuned Models
>
> We respectfully disagree that this is a workaround. We show with detailed analyses in Section 3 that base models are more diverse than instruction tuned models and instruction tuned models are more accurate. We leverage both these properties into a single pipeline, BARE. Further, our experiments show that relying solely on one type of model is limiting, indicating that leveraging both types of models is necessary for high-quality data generation and not just a workaround.
>
> ### Clarifications
> Regarding our claim on being the first work to show the value of base models in synthetic data generation. The arXiv work the reviewer mentions does not compare against a comparable pipeline utilizing solely instruction-tuned models. BARE, on the other hand, explicitly demonstrates how utilizing base models in synthetic data generation can be superior to relying solely on instruction-tuned models, which is a stronger demonstration of value. However, we will mention the work and clarify our claim in our revision.
>
> Regarding the number of samples used by Humpback [d], while they “only” use 3.2k labeled samples (compared to our 3), they do additionally require 500k unlabeled snippets (as indicated in their Section 3.1). We will clarify this in our revision.
>
> Regarding “generations derailed” for GSM8K when generating with Instruction-tuned models only, under this setting the instruction-tuned model failed to generate problems matching the specified format, leading to malformed examples we could not train with. To maintain parity with other single-model methods, we did not make additional model calls to repair the examples.
>
> Regarding the IR metric, we acknowledge that it is not a perfect metric. However, for our purposes we found that it gave sufficient intuition to motivate our design of BARE. Obviously synthetic examples (like the base generation in Figure 5) would be easily identifiable by models like GPT-4o. We believe that our setup makes IR a reasonable, though not perfect, proxy for entry-wise quality/realism. The more important metric for overall dataset quality is downstream performance, which is our primary metric in our evaluations in Section 5.
>
> ### References
>
> [a] The Unlocking Spell on Base LLMs: Rethinking Alignment via In-Context Learning. ICLR 2024.
>
> [b] ALMA: Alignment with Minimal Annotation. arXiv 2024.
>
> [c] Is In-Context Learning Sufficient for Instruction Following in LLMs? ICLR 2025.
>
> [d] Self-Alignment with Instruction Backtranslation. ICLR 2024.

---

### Meta-Review · Area_Chair_WBdk · 2026-01-13

**Summary:**

This paper proposes a two-stage pipeline of data synthesis. Based on the observation that base models with few show prompting can generate diverse data but of lower quality, while instruct models tend to generate higher quality data but of lower diversity, this work proposed to first diverse synthesize data using the base model, and then use an instruct model to refine such generated data. Experiments verify that the generated data from base models is more diverse, and also that the data can be used for finetuning models.

Overall, reviewers share concerns about the novelty of the approach and the lack of comparison to baselines in the literature. While the authors address some reviewer concerns, it seems unlikely that reviewers will flip their decisions if they were given the chance to update their reviews. Therefore, I'm not recommending the acceptance of this paper in this case.

**Reviewer Concerns:**

1. The concerns about similarity to existing works are partly addressed by clarifying the differences in this work. However, the existence of a large body of works that show philosophically similar insights still undermines the novelty of this work.
2. The concerns about the small size of the generated dataset if addressed by clarifying that the appendix includes a larger scale experiment. I highly recommend the authors to move this to the main paper in their next version.
3. The reliance on two models of both a base model and an instruct model seems not fully addressed despite clarification from the authors.
4. The concern of why LoRA is used in experiments is justified by the recent thinking machines blog by John Schulman, but that blogpost is more focused on RL setting so I think this concern is not fully addressed.

**Reviewer Scores:**

1. C12M: the concern on the novelty in the context of related works is partly addressed, but it seems that the major concern of the novelty of this work given the large body of similar works is not fully addressed. Therefore, I think it is likely that this reviewer will bump up their score to 3 or keep their score.
2. Qc13: the concern of why LoRA is used is not fully addressed as detailed above. Therefore, I think this reviewer will likely keep their score.
3. mjgC: the concern of the dependence on refiner model is partly addressed as the authors show that using weaker refiners also works. However, I think this reviewer's main concern is about novelty and this reviewer is likely to keep their score.
4. gmgL: similarly, due to the concern about novelty, this reviewer is likely to keep their score.

---

### Decision · Program_Chairs · 2026-01-26

Reject